# Boundaries steer the contraction of active gels

Matthias Schuppler[1], Felix C. Keber[1], Martin Kröger[2] & Andreas R. Bausch[1]

Cells set up contractile actin arrays to drive various shape changes and to exert forces to their environment. To understand their assembly process, we present here a reconstituted contractile system, comprising F-actin and myosin II filaments, where we can control the local activation of myosin by light. By stimulating different symmetries, we show that the force balancing at the boundaries determine the shape changes as well as the dynamics of the global contraction. Spatially anisotropic attachment of initially isotropic networks leads to a self-organization of highly aligned contractile fibres, being reminiscent of the order formation in muscles or stress fibres. The observed shape changes and dynamics are fully recovered by a minimal physical model.

[1] Department of Physics, Cellular Biophysics E27, Technical University of Munich, Garching D–85748, Germany. [2] Department of Materials, Polymer Physics, ETH Zurich, Zurich CH–8093, Switzerland. Correspondence and requests for materials should be addressed to M.K. (email: mk@mat.ethz.ch) or to A.R.B. (email: abausch@mytum.de).

Cells and tissues are driven far from thermodynamic equilibrium by mechano-chemical stresses. Cellular forces, generated by actomyosin contractions, are essential for 'internal' processes like controlling elastic moduli and shape stability of the cytoskeleton but also to respond to 'external' stimuli, such as sensing properties of their surroundings and adapting to them[1–5]. Depending on the specific physiological task, the contractile arrays exhibit different shapes and are embedded in different environments. For instance, in cytokinesis contractile stresses are confined to the equatorial plane to drive furrow ingression whereas during cell migration the cell shape symmetry is broken to establish a front-back polarity. In all these processes the contractile arrays are embedded and constantly attached within their various biological environments. To this end little is known about the interplay between initial assembly of the contractile structures and how this affects the cellular dynamics and structure formation. Clearly, in cells besides the biochemical signalling the attachment of the contractile structures to adhesion sites or the anchoring within the cortex are key for their assembly and function, yet it is unclear if such boundary conditions would be already sufficient to drive the formation of contractile cellular modules[3,6].

To shed light on these questions, the concept of active gels has been developed[7]. These systems, unlike structure formation in passive out-of-equilibrium systems which are formed by nucleation and growth mechanisms, rely on force generation and dissipation in a viscoelastic environment.

A minimal system of a dynamic cytoskeletal network helps to understand the physical principles of the macroscopic contraction as a consequence of the interaction between actin, crosslinks and myosin II (refs 8–11). The intricate interplay between these constituents give rise to a rich phase behaviour[10,12–14]. Depending on the parameters, either a dynamic steady state, with reminiscent nucleation and growth mechanism of clusters can be observed, or in a case of a stable percolating network a collapse can be sensitively tuned by the motor concentrations[12,14].

Here we demonstrate that the contractility and structure formation of active actomyosin systems is steered by their boundary conditions. By independently tuning the activity and the shape of active gels by light, we show that the force balance at the boundaries is responsible for the observed shape transformations and their dynamics. Shape transformations are modelled by a static spring network model, while the observed dynamics are fully recovered by a minimal set of critically damped harmonic springs. Anisotropic attachment of the network leads to the formation of highly aligned contractile fibres, reminiscent of contractile modules found in cells. The activity of the myosin is controlled by the addition of blebbistatin and its inactivation by light.

## Results

**Light-stimulated contraction of active actin gels**. We assemble isotropic gels of actin filaments and myosin II in the presence of blebbistatin. This myosin II inhibitor arrests the motor proteins in a weakly actin-attached state with strongly quenched ATPase activity[15]. Illumination with blue light (488 nm) inactivates blebbistatin and consequently activates the myosin. Thereby, we find that no further competitive inhibition of the once activated motor is possible by the remaining freely diffusing active blebbistatin in the surrounding. Thus, the sustained activity of the illuminated area over a time scale of an hour indicates that the inactivated blebbistatin remains covalently attached at the myosin heads[16].

We use a scanning confocal spot to activate (stimulate) different two-dimensional geometries of the networks, which remain attached to a surrounding inactive gel (Fig. 1a) for up to 160 repetitive stimulation cycles, each of which lasts 640 ms. Subsequent imaging of the actin network was performed, by labelling the actin with Alexa 647N phalloidin, which is excited with a 633 nm laser line. The laser light-induced myosin activity results in a sustained contraction of the gel over the course of up to more than 90 min without any further stimulation. The contraction is sensitive to both the shape and size of the activated area, and the intensity and duration of light exposure. Where a circular area is activated, the gel contracts in a shape preserving manner towards its centre of mass (Fig. 1b, first row). The observable projected area of the active gel continuously decreases (Supplementary Movie 1). The contraction terminates when either the maximal packing density of the actin is reached or the elasticity of the gel is too high for the motors present. Here, forces occur only normal to the active-inactive interface, while tangential forces are equally balanced due to the circular geometry of the activated area.

**Effect of boundaries on the contraction of active actin gels**. This reasoning implies that the shape would not be preserved in the case of activated circular asymmetric areas. In such cases tangential force imbalances would occur which in turn lead to asymmetric deformations altering the geometry and line curvature of the active gel while contracting. Indeed, if quadratic areas are activated, their shapes alter during contraction (Fig. 1b, second row). For every point along the interfacial line the resulting force is set by the locally active and passive environment. If for adjacent points this resulting force has the same magnitude and direction with respect to the interface normal, this structural element does not give rise to a shape change. At the corners the force imbalance along the interfacial line gets changed abruptly. The symmetry at the corners of the square thus results in a different pulling direction compared with the edges of the square. As a consequence, a gradient of force imbalances along the boundary emerges. On its way from initial to final state, all intermediate states exhibit different curvatures of their boundaries with slightly altered force imbalances along their boundaries. These intermediates can be regarded as new initial states giving rise in turn to a further shape change. For full squares the inward movement for the width in the middle is higher than for corners and this results in a concave deformation of the interfacial line (Fig. 1b, second row).

By activating a thin hollow square a second interface to the passive environment is created. For the outer boundary, this geometry is identical to the full square and should give rise to comparable shape transformations. But the force imbalances at the outer boundary have to be superimposed with the ones of the inner boundary. There, inner corners are geometrically exposed points since they experience contractile forces from 270°, whereas neighbouring points along the straight boundary experience forces from 180°. With this both the magnitude and the direction of force acting on neighbouring structures change, creating a steep gradient of force imbalance along the boundary. As a result, the inner corners contract more than the outer corners which are delayed. At the centre of the four long inner and outer edges the force balance does not change for the neighbouring points. Consequently, the curvature along the inner boundary tends to assimilate and the overall shape change exhibits a convex transformation (Fig. 1b, third row). Thereby, boundary-induced shape transformations seem to get annihilated inside activated patterns with increasing distance to boundary.

If we activate a thick hollow square, we observe both features of the full and the thin hollow square. The inner boundary changes shape towards a circular geometry while the corners are

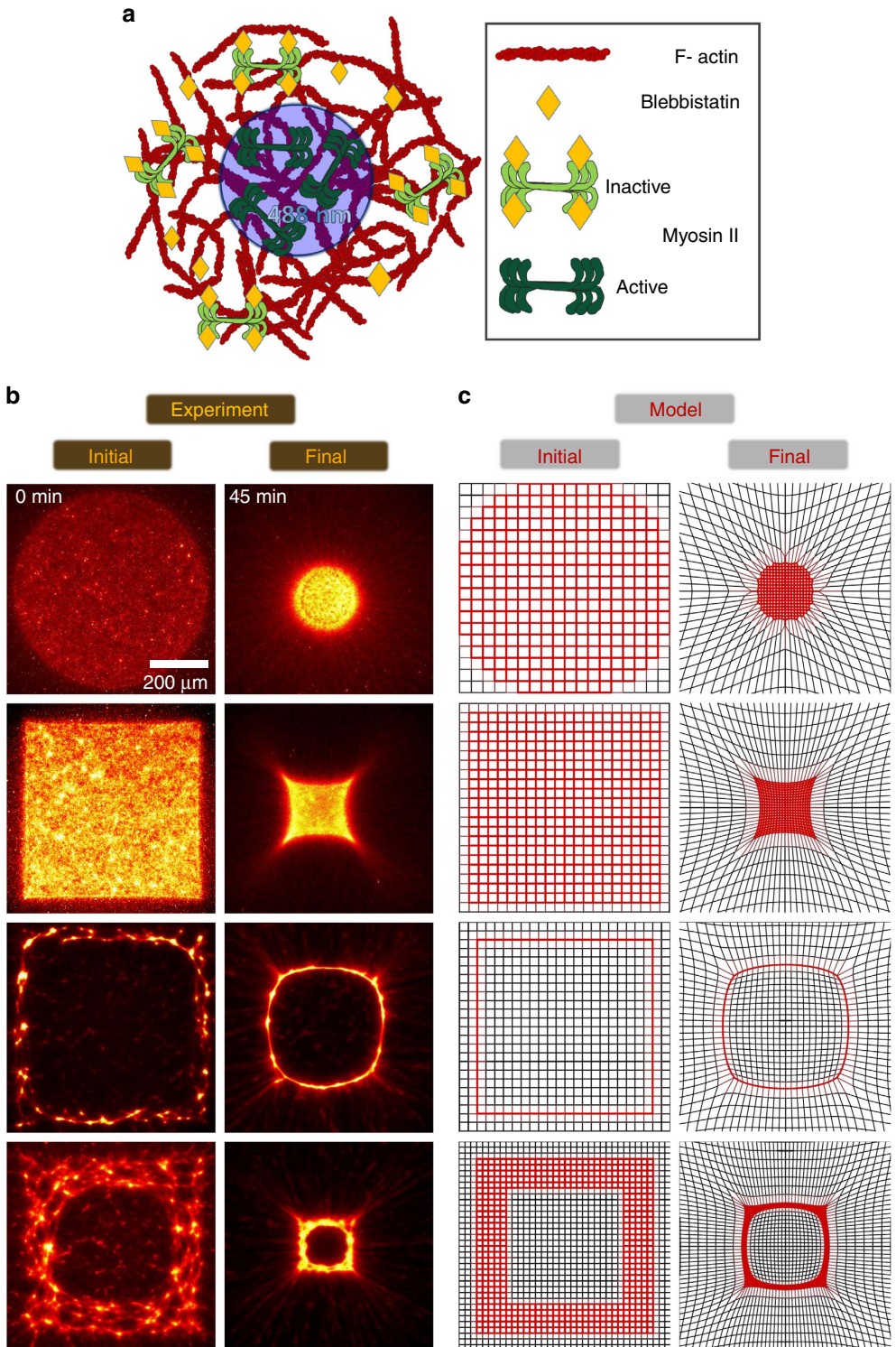

**Figure 1 | Global shape changes of active actin gels.** (**a**) At the beginning of the experiment myosin II filaments are activated by scanning patterns with a confocal blue (488 nm) argon laser beam for $n = 20$ consecutive cycles (640 ms for one cycle), thereby inactivating the blebbistatin. The activation of the myosin is stable for at least 90 min and no further activation is necessary during the further contraction. This observed permanent activation is attributed to the light-induced covalent modification of the myosin II by the blebbistatin. (**b**) Start ($t_{init} = 0$ min) and end ($t = 45$ min) configuration of the fluorescently (phalloidin-Alexa 647N) labelled actin network, imaged with a 633 nm HeNe laser line. During the stimulation process ( <1 min) activated molecular motors already accumulate fluorescent actin from all directions leading to an increased fluorescence directly after stimulation is finished. Circular patterns contract shape preserving whereas various initial geometries with corners entail non-affine deformations during contraction. Scale bar is identical for all figures. (**c**) Corresponding results of our static spring network model (Supplementary Note 1 and Supplementary *Mathematica* notebook). The initial state is set by the activated region, the final state is obtained by efficient inversion of a sparse matrix with a suitable value for the ratio of spring coefficients $K = 0.13$ on a $30 \times 30$ square grid for all panels.

surrounded by concavely transformed boundaries (Fig. 1b, fourth row).

**Static spring network model recovers shape deformations.** The described observations give rise to a simple model representing the passive and active parts of the gel by crosslinked non-tearable springs of different stiffness, that are brought by illumination into a highly non-equilibrium initial state. Because the conformations of actin filaments can be considered relaxed in the initial state, harmonic springs (polymeric dumbbells) will likely be sufficient to recover experimental results. Initially our model system consists of a regular two-dimensional array of crosslinker nodes contained in a periodic simulation cell that is sufficiently large compared with the activated area. All crosslinkers are permanently connected to all of their initially four nearest neighbours by identical weak harmonic springs with vanishing rest length and spring coefficient $k_w$. Reflecting the geometry of the activated region, the spring coefficients are instantaneously set to an activated value $k_s > k_w$. Thus, motor activity is described by the driven redistribution of elastic energy stored in connected springs of different spring coefficients, but identical initial elongation. Because the forces are linear in the displacements, a sparse matrix $\mathbf{A}$, that is given by the topology of the initial grid and the choice of the activated area, and a constant vector $\mathbf{b}$ determined by the boundary conditions fully describe the transition from the initial configuration vector $\mathbf{x}_{init}$ (Fig. 1c, left column) to the force-free, final state via $\mathbf{x}_{fin} = \mathbf{A}^{-1} \cdot \mathbf{b}$ (Fig. 1c, right column and Supplementary Note 1).

For any geometry/distribution of activated and strong springs the observed shape changes can be retrieved with this static spring network model, and the results are independent of the simulation cell size in the thermodynamic limit. While the initial shape change of the interface is determined by the nearest surrounding and thus identical all over the interface for circular or infinitely flat geometries (circles and straight lines), the spatially varying effective spring coefficients as they occur for less ideal activated regions dominate the relaxation behaviour. The force field produced immediately after activation already determines the final, relaxed and force-free geometry of the activated region, because we consider permanent, harmonic springs, and because this initial force field is also reflected by the matrix $\mathbf{A}$.

The final state provided by the model is given by the ratio $K = k_w/k_s$. Consequently, for smaller $K$, the contraction ratio should be smaller. Indeed, in the experiment we observe that by increasing the number of stimulation cycles the density of active motors, corresponding to the spring constant $k_s$ gets increased and circular patterns of constant initial circle radius $C_{init}$ get contracted to a smaller final area, characterized by its final radius $C_{fin}$ (red circles in Fig. 2a). The static spring network model suggests $C_{fin}/C_{init} = K$, while $K$ is expected to depend on the number $n$ of stimulation cycles as $K(n) = 1 + (K_\infty - 1)[1 - (1 - P)^n]$ (red solid line in Fig. 2a), where $P \in [0,1]$ represents the stimulation efficiency, and $K_\infty = \lim_{n \to \infty} K(n)$, provided that activated material cannot be stimulated twice (Supplementary Note 2). This simple static model fits the experimental data very well and allows us to extract the stimulation efficiency $P$ and saturated ratio $K_\infty$ (reported in Fig. 2a) and to predict the final shape as function of the number of stimulations. We find for ratios of spring coefficients $K$ values in the range between 1/6 and 1/4 for $n = 20$ stimulation cycles (Fig. 2b). This corresponds to ratios between elastic moduli of the quiescent and stimulated static spring network. For the experimental system this ratio captures both, the activity of the motors and the resulting elastic response of the active gel region[17].

**Speed of contraction depends on the distance to the boundary.** As the constituents of our gel are subjected to friction the transition from initial to final state happens at finite velocities. The magnitude of these should be affected by the density of the molecular motors. Indeed, with increasing density of active motors we observe that also the maximal contraction speed $v_c^{max}$ of an activated circle with initial diameter of 775 µm, measured from time-dependent positions of opposing boundaries, gets increased (black squares in Fig. 2a), as the force acting on the boundary (force per perimeter) gets increased. When all available motors are activated, further stimulation does not affect neither $v_c^{max}$ nor the contraction ratio $C_{fin}/C_{init}$ (Fig. 2a).

The static spring network model (Supplementary Note 1) suggests that proportional to the number of sequentially arranged strong springs (contractile units) the force on the boundary of the activated pattern gets increased. A higher force would imply a higher maximal contraction speed $v_c^{max}$. Indeed, if we increase the linear size of symmetric patterns, for example, circles and squares, from 0 to 700 µm, the maximal velocity of the boundary $v_c^{max}$ increases linearly with the diameter for circles, and the minimum horizontal width for squares (Fig. 2b). As a conclusion we find that $v_c^{max}$ is governed by the number of contractile units per length and the velocity of contraction of each unit cell[18].

For our systems, the maximal contraction speed of the boundary $v_c^{max}$ is not reached instantaneously upon activation. Measuring local contraction rates (spatial velocity gradients) by particle image velocimetry (PIV) analysis (see Methods section) reveals that after activation contraction is taking place only close to the boundary. Further away from the boundary the contraction lags behind (Fig. 2c–d). In the centre of the activated pattern, the local contraction rate starts off at almost zero, indicating hardly any relative, contractile displacement of structures. During contraction the local rates get increased to the initial value measured at the boundary. This value is reached earlier for structures that are closer to the boundary. This observation can be rationalized by considering the local force balances. In the initial state, right after activation, the active gel is isotropic and local active forces are locally balanced. This symmetry is only broken at the boundary. Thus, motors in close proximity to the interface translate the boundary because of this force imbalance. Thereby the force imbalance acting on the structures located at the interface emerge from the immediate active and passive environment. Forces generated further away from the boundary do not contribute to a net movement of the boundary, but rather increase the level of tensile stresses on their neighbours. In turn, this continuous auto-induced increase of tension allows motors to exert forces across larger regions. Once the radius of this interaction region exceeds the distance to the boundary, motors sense the presence of the boundary. They are now able to translate their forces into a displacement of the boundary and participate in contraction. Thus, the information of unbalanced forces, starting from the boundary, penetrates the network during the onset of contraction. The maximal velocity of the boundaries is reached when all contractile units between them sense their presence. From tracing network inhomogeneities or fiducial tracer particles, we can conclude that the network contracts by reducing the relative distances without positional redistributions of material inside the network. The increase of the speed gradient correlates thereby with the densification of the network.

**Temporal asymmetric contraction of rectangles.** The necessity of force imbalance for the contribution of material to the contraction speed implies that the contraction process depends on the exact asymmetry of the activated areas. For asymmetric patterns, for example, rectangles of initial aspect ratio $\alpha_{init} \equiv X_{init}/Y_{init} > 1$ (see Supplementary Movie 2), the distance of

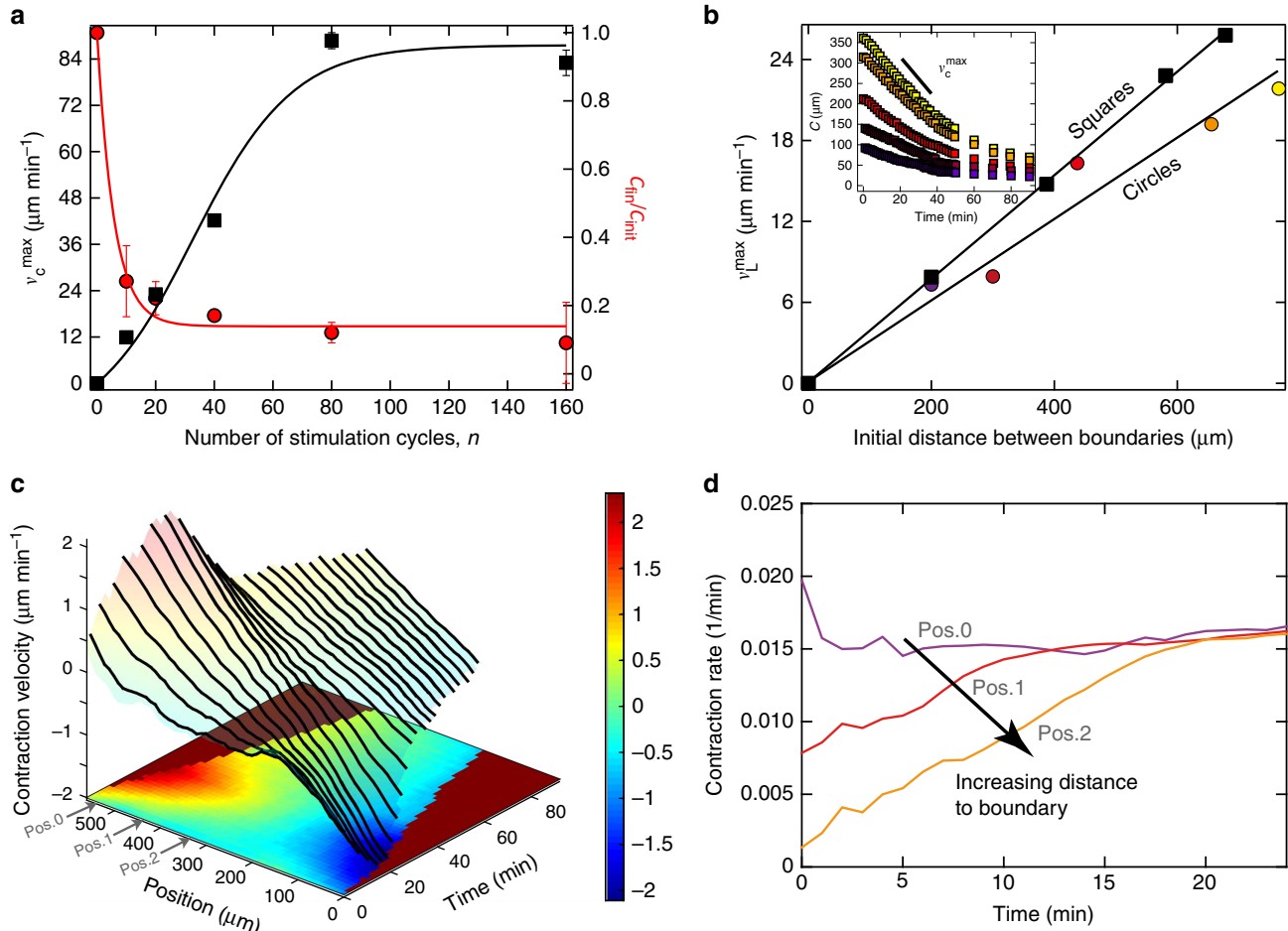

**Figure 2 | Contraction speed depends on geometry. (a)** Final contraction ratio $C_{fin}/C_{init}$ (red) and maximal contraction velocity $v_c^{max}$ (black) of initially identical circles (radius $C_{init} = 388\,\mu m$) depends on the number of stimulation cycles $n$, corresponding to the density of active motors. Solid lines represent our calculations quantified by the ratio of spring coefficients for which we find $K = C_{fin}/C_{init}$ (Supplementary Equation 1 with $P = 11\%$ and $K_\infty = 0.15$) and $v_c^{max}$ (Supplementary Equation 19), both as a function of $n$. **(b)** Maximum contraction velocity versus the initial distance between boundaries for symmetric patterns (diameter for circles denoted as coloured circles and the width in the middle for squares denoted as black squares) at identical $n = 20$. Solid lines represent linear fits in accordance with Supplementary Equation 15 of our minimal dynamic model, where $v_L^{max} \sim L_{init}$ is proportional to the initial distance between boundaries. The inset shows the radii of activated circles as a function of time for the corresponding colour coded circles. **(c)** Internal dynamics during the onset of contraction of a full square (half-width $X_{init} = 290\,\mu m$) after $n = 5$ stimulation cycles. Colour coded are the velocities of contractile structures along the $x$-coordinate obtained by PIV (Supplementary Fig. 4). The boundary velocities increase to their temporal maxima in the course of time before approaching equilibrium in the long term. To visualize the symmetry and the size reduction of the contraction we show the $z$-projection of the corresponding three-dimensional plot. **(d)** To determine where and when motor activity results in a contraction, the local velocity gradients $dv_x/dx$ for three regions (Pos.0, Pos.1 and Pos.2) with increasing distance to one of the interfaces (0 μm, 100 μm and 200 μm) are calculated from the data in **c** (indicated by the grey arrows, region width is 30 μm) and plotted over time. Close to the interface the contraction rate is high after activation and stays constant. With increasing distance to the boundary the contraction rate is close to zero and gets increased to the contraction rate measured for the boundary. Hence, motors initially contract structures only at the interface and the global ability to contract penetrates the activated area from there. We find that the maximal velocity of the boundary is observed when the centre region exhibits the initial boundary contraction rate (here at $t \approx 16$–20 min).

the centre region to the boundary is different for the two principal axes. Consequently, the maximum contraction velocity for the long axis, $v_X^{max}$ should be reached later than for the shorter axis, $v_Y^{max}$ (Fig. 3a and Supplementary Fig. 2). There will exist regions in space that have the ability to contract in $y$-direction while still being unable to contract in $x$-direction. Yet the initial contraction velocity around the perimeter of the pattern should be uniform, independent of the geometry. Which is indeed what we see (inset of Fig. 3b). As we observed identical aspect ratios before and after contraction (red squares in Fig. 3b), this implies a temporal asymmetric contraction, where the short side contracts before the long side, followed by the contraction of the long side. This asymmetry in time gets captured in a transient increase of the aspect ratio to its maximum $\alpha_{max}$ (blue circles in Fig. 3b).

As we have seen for symmetric patterns the maximum contraction speed is proportional to the initial distance between the boundaries. For asymmetric patterns with identical areas, this would predict that the ratio $v_X^{max}/v_Y^{max}$ of the maximal velocities of the principal axes equals the initial aspect ratio. However, experimentally we find $v_X^{max}/v_Y^{max} = \alpha_{init}^{1-\gamma}$ with $\gamma = 0.4 \pm 0.1$ (Fig. 3c). This non-trivial dependence indicates an intricate dynamic coupling of the principal axes, resulting in either a hampered contraction along the length or an enhanced contraction along the width.

To shed light on this issue we activate a series of rectangles with constant half-length $X_{init}$ and variable half-width $Y_{init}$. The maximal velocity of the width increases linearly with increasing width as expected for the symmetric, undisturbed contraction,

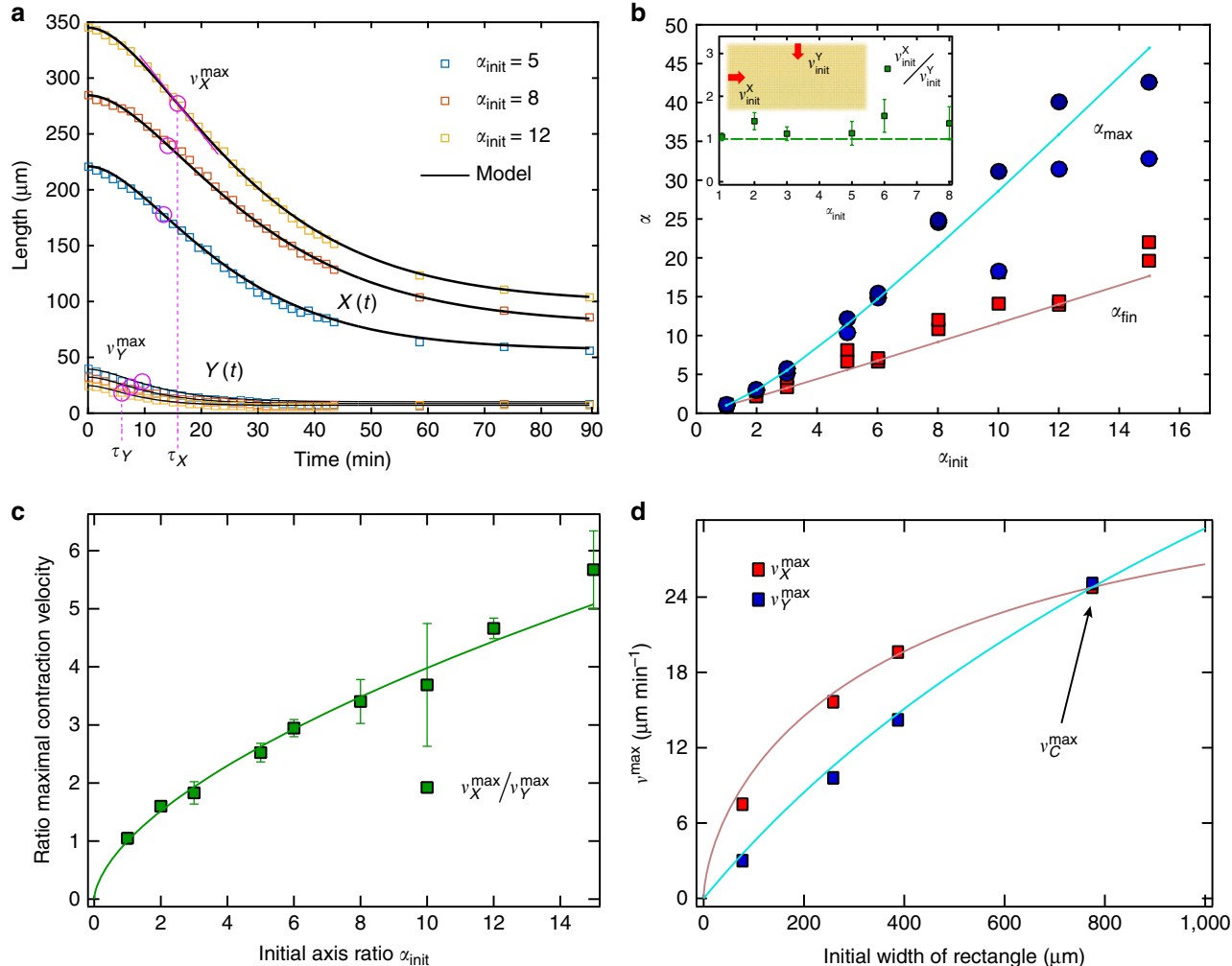

**Figure 3 | Dynamics of contraction is set by shortest axis. (a)** Experimentally measured long axis $X(t)$ and short axis $Y(t)$ for filled rectangles with different initial axis ratio $\alpha_{init}$, with constant areas $X_{init}Y_{init} = 10,000\,\mu m^2$, together with Supplementary Equation 3 (solid lines), where $K = X_{fin}/X_{init} = Y_{fin}/Y_{init}$ and $\tau_X$ and $\tau_Y$ had been determined from the fits. While $K$ is quasi-constant due to identical stimulation density, the $\tau$'s depend on axis ratio (Supplementary Fig. 3). For each dimension $L$, the maximum velocity $v_L^{max}$ occurs at the compression time $\tau_L$ that simultaneously determines the long-time relaxation behaviour (Supplementary Equation 15). The contraction along the long axis $X(t)$ exhibits a strong delay compared with the contraction of the short axis $Y(t)$. The maximal contraction speed occurs much earlier in $Y(t)$ as shown in Supplementary Fig. 3. **(b)** Characteristics of asymmetric contraction of initially asymmetric geometries: $\alpha_{fin} = X_{fin}/Y_{fin}$ (red squares) and $\alpha_{max}$ (blue circles) both as a function of $\alpha_{init} = X_{init}/Y_{init}$, shown together with results of our minimal dynamic model (Supplementary Notes 3). Inset: The ratio of velocities immediately after stimulation of the perpendicular interfaces is independent of the geometry. Because the velocities vanish at equilibrium, a limiting ratio of unity signals that there is a unique acceleration and unique force strength normal to initially flat interfaces, as explained in Supplementary Notes 3. **(c)** Ratio of the maximal velocities obeys the relation $v_X^{max}/v_Y^{max} = \alpha_{init}^{1-\gamma}$ with $\gamma = 0.4$. **(d)** For rectangles of constant half-length $X_{init} = 400\,\mu m$ and variable width, $v_Y^{max}$ (diamonds) depends basically linearly on $Y_{init}$ as expected for the undisturbed contraction, whereas $v_X^{max}$ (squares) gets increased nonlinearly with $Y_{init}$, indicating a hampered contraction along the long side. Using the $v_c^{max} \approx 24\,\mu m\,min^{-1}$ value we read off for the square case. Supplementary Equation 20 of our minimal dynamic model predicts $v_X^{max} = v_c^{max}/\alpha_{init}^{\gamma/2}$ (red line) and $v_Y^{max} = v_X^{max}/\alpha_{init}^{1-\gamma}$ (blue line) with $\gamma = 0.4$, in agreement with these data. All network results shown here are identically stimulated with $n = 20$ cycles.

whereas the maximum contraction speed along the length gets increased nonlinearly as the initial width gets larger (Fig. 3d). If there was no coupling, the maximal contraction speed of the length should be independent of the other dimension's size. Yet, the shortest side of an asymmetrically shaped, isotropic, contracting gel determines the overall dynamics of contraction.

**Critically damped springs model recovers contraction dynamic.** As all the dynamics of the length changes of the contraction are reminiscent of a critically damped oscillator (Fig. 3a), we can readily model the observed dynamics of the interface by replacing the large grid of springs by a few effective (coarse-grained)

critically damped springs in a minimal dynamic model (Supplementary Note 3). The static spring network model and the minimal dynamic model are independent. We only impose that the stationary state of the minimal dynamic model must match the static spring network result for simple geometries. Both models are linked by the ratio of spring constants $K$, which determines in both models the ratio between initial and final sizes. Since $v_c^{max}$ is found to be proportional to the length of the symmetric structures or circles and squares $C_{init}$, the compression time $\tau_c$, at which the maximal velocity is reached, is size-independent. This is a consequence of critically damped dynamics. We extract $\tau_c \approx 12\,min$ using the fitted values reported

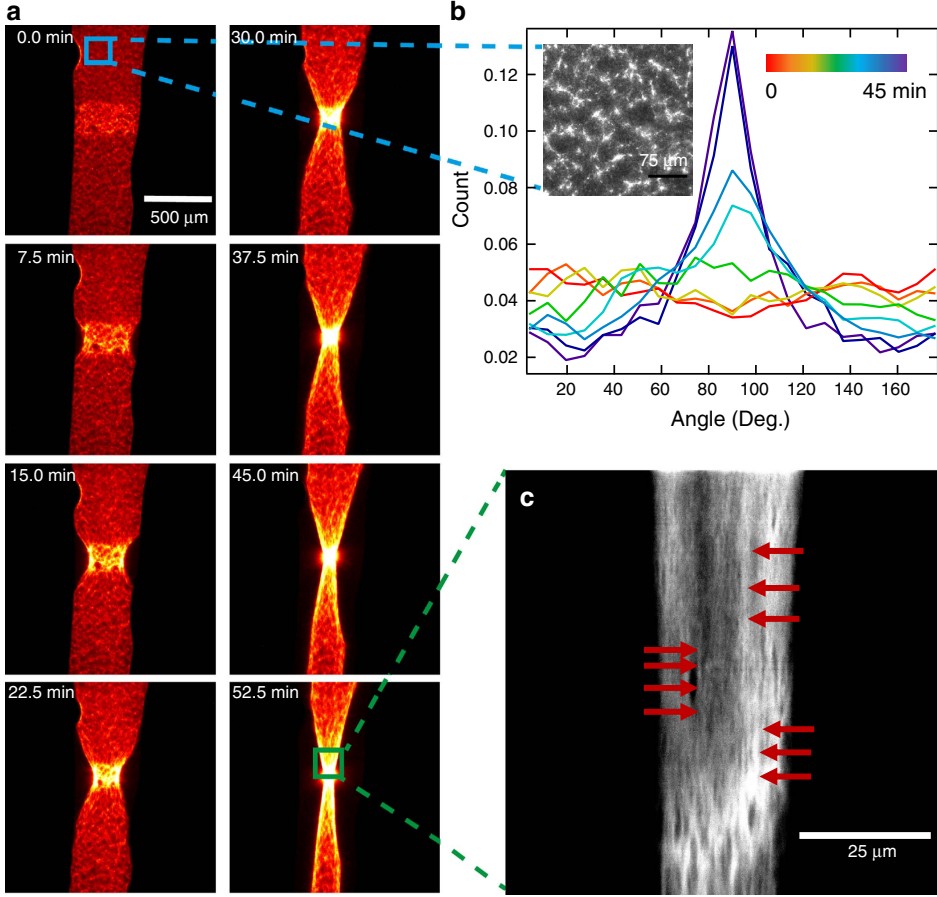

**Figure 4 | Asymmetry-induced formation of contractile fibres.** (**a**) Time series of a contracting array with a free boundary perpendicular to the channel axis. The stimulated area ($n = 20$ cycles) has an initial size of $\approx 100{,}000\,\mu m^2$. (**b**) Analysis of the whole images reveals the emerging orientation of actin structures parallel to the channel axis at 90°. Orientational distributions are shown for the 8 to **a** corresponding (colour coded) time points during the contraction. Inset: zoom into a yet uncontracted region reveals an apparently isotropic network, in agreement with the distribution at the starting point of stimulation (red line). (**c**) Zoom in after completed contraction: Red arrows denote increased alignment of actin structures as the result of motor activity and confinement of tension, quantitatively reflected by the peaked distribution (dark blue line).

in Supplementary Fig. 3. This compression time $\tau_c$ is determined by the densification and concomitant increase of the modulus of the gel and the finite speed and stall force of myosin II.

For an asymmetric pattern like the rectangle, both axes feel the same initial normal force, resulting in the same initial speed. As in addition the starting and ending ratios of the axes are equal, the springs of both principal axes are coupled by the initial and final states. By these observations, the minimal dynamic model readily fits the dynamics of both axes $X(t)$ and $Y(t)$. The same scaling of the ratio of the maximal speeds $v_X^{max}/v_Y^{max} = \alpha_{init}^{1-\gamma}$ is obtained with $\gamma = 0.4$. All fits shown in Figs 2a,b and 3 are obtained by this simple, coupled differential equation using one global set of parameters compatible with $K \approx 0.23$ and $\tau_c \approx 12$ min for $n = 20$ stimulation cycles (Supplementary Fig. 3).

For all linear dimensions $L \in \{C,X,Y\}$ this minimal dynamic model predicts that the compression time $\tau_L$ determines both the long-time exponential decay of $L(t)$ and a maximum velocity $v_L^{max} \sim (1 - K)L_{init}/\tau_L$ that actually occurs at the compression time $t = \tau_L$. The compression times for the axes of a rectangle are identical to $\tau_c$, times a factor that depends on the axis ratio alone. Symmetry reasons together with the above ratio of maximal velocities suggest $\tau_X = \tau_c \alpha_{init}^{\gamma/2}$ and $\tau_Y = \tau_c \alpha_{init}^{-\gamma/2}$, in agreement with our data for fixed $X_{init}$ (Fig. 3d and Supplementary Equation 20), and the known ratio of maximal speeds.

A spring as part of the active network can be considered to represent the mechanical action of an individual motor filament

that exerts polar forces. The motors can only be engaged into the direction towards the nearest force-unbalanced boundary. The observed critically damped dynamics of the shape changes may arise from large algebraic connectivity of the network (Supplementary Note 5), or time-dependent friction and spring coefficients within a minimal dynamic model. We did not investigate the question, under which conditions the static network spring model, whose equilibrated state is insensitive to inertia and friction effects, produces a critically damped dynamics of the interface[19]. Because of the appearance of critical damping, the friction and spring coefficient of the quiescent model gel are inherently coupled to allow for a maximally efficient relaxation. This is a very remarkable property our system seems to share with several natural interaction networks[20–22], where a critical damping is essential.

The introduction of a nonlinear spring response, which could address the gel densification can also lead to such a coupling, but would lead to a significantly increased number of free fitting parameters. The microscopic interpretation of the parameters $K$ and $\tau_c$ goes beyond the scope of the present work and will need to include a realistic incorporation of the myosin filaments and the local elasticities of the network.

**Initial anisotropic attachment entails alignment of fibres.** So far we have shown that for both the contractile dynamics as well as the shape changes the interplay of the contractile, active

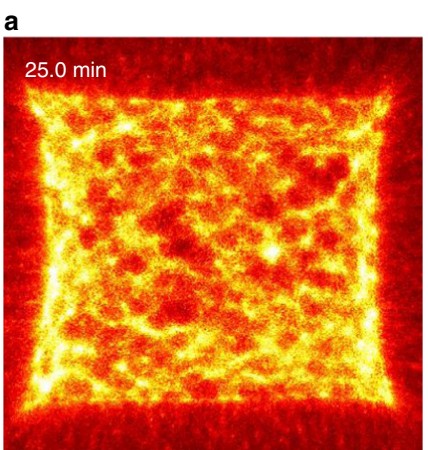
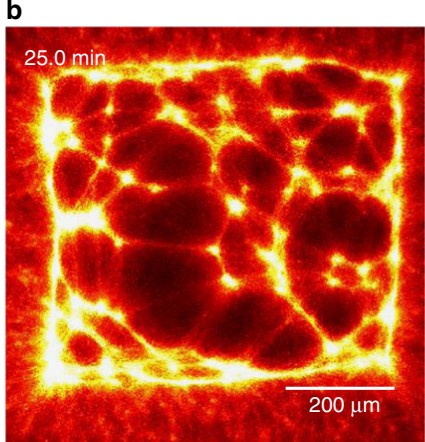

**Figure 5 | Initial connectivity is essential for predictability of shape changes.** Upon decreasing the (**a**) initial average F-actin length by the (**b**) addition of 2 nM capping protein the connectivity of the network is decreased. The fluorescence of the actin network is shown in the images. The gel breaks up into disjoint clusters in **b** while contracting non-uniformly. Both patterns are of identical size ($\approx 600,000\ \mu m^2$) and activated during an identical amount of $n = 20$ stimulation cycles.

with the surrounding viscoelastic, inactive array is essential. By generating a free surface at the walls of a microfluidics system we break the symmetry of attachment for the activated area. No restoring forces exist for contractions perpendicular to the channel axis after the negligible, initial detachment of the gel from the channel walls. Thus the active gel contracts first predominantly radially leading to an alignment of the filaments in the activated area along the axis of the channel (Fig. 4a). The strain fields produced by the contractile area penetrate deep into the inactive network. In this geometry, the build-up of tension is restricted to the long axis of the channel. Thus an increased order parallel to the direction of tension is observed (Fig. 4c). We extracted the distribution of fibre orientations at eight equidistantly spaced time points during the contraction within the channel setup (Fig. 4b), as described in the Methods section. One can clearly recognize an isotropic distribution of the fibres at the beginning of the experiment and a developing peak at 90°, corresponding to alignment preferentially along the main channel axis.

The alignment of cytoskeletal structures parallel to a free boundary is the result of the initial anisotropic attachment in concert with the large nonlinear deformations induced by the actomyosin contraction. Hence, an initial anisotropy only present at the boundaries translates into a characteristic feature of the entire gel. Fibre alignment can thus be seen as a direct consequence of isotropic contraction of one-dimensionally attached gels. Such boundary driven plastic deformations might serve as directional template for biochemical and further structural polarization. This mechanical response may elucidate complex cell behaviours observed in assembly of stress fibres or myofibrillogenesis, where large deformations and mechanical tension have been shown crucial[23].

**Connectivity of the network is prerequisite for contraction.** A prerequisite for the observability of shape transformations in general is the integrity of the connections between neighbouring active and passive structures (Fig. 5, left panel). If motor filaments create ruptures inside activated areas, the predictability of shape changes breaks down. A decrease of the average actin filament length is achieved by adding 2 nM capping protein[24].

After motors are switched to an active state predominantly small clusters of actin filaments are formed spontaneously (Fig. 5, right panel). Because of this emerging inhomogeneity of the actin density, the forces inside the active structures are no longer balanced. Once formed, the small clusters fuse and this formation of a few large clusters results in a decline of the network's connectivity. The more actin is accumulated in large clusters, the less material is available to connect different clusters with each other. During this process motors generate many concurrent ruptures that constantly redefine the boundaries within the active pattern, ultimately resulting in a distribution of disjoint clusters and leading to a collapse of network structure[14]. Such inhomogeneous rupture events can be readily accommodated into our network model by including short-ranged attractive interactions, which results in a so-called elastic Lennard–Jones network[25,26]. The equilibrated state of such networks bears remarkable similarities with the shown experimental data.

Our findings hint at a generic mechanism for the built up of structures as a consequence of the initial shape and boundary conditions. It has been shown before that the intricate interplay between shape and tension accounts for various spatial organizations on the cell and tissue level[2,27–29]. The presented mechanism emphasizes the influence of geometry on irreversible self-organized shape changes in the micro-structure of subcellular constituents without any additional regulation of the motor-driven activity. Breaking a symmetry at the boundaries by introducing focal adhesion may be the sufficient clue for the self-organized structure formation of stress fibres. Clearly, the presented system is static in nature, and the dynamic turnover cellular structures will significantly alter the exact mechanisms of their emergence—yet the symmetry breaking of the force balances at the boundary seem to be an important prerequisite. The role and importance of boundary conditions is generic and may needed to be revisited in the case of larger length scales, leading for example to the structure formation in tissues. This work contributes to distinguish between purely mechanical self-organization of contractile gels within cells and adaptive biochemical responses to mechanical signals.

## Methods

**Protein purification.** G-actin[30,31] and myosin[32] were extracted from rabbit skeletal muscle by standardized protocols. Actin was fluorescently labelled with alexa 647 N phalloidin (Invitrogen) at a label degree of 2%.

**Sample preparation and optical stimulation.** In the presence of 0.1 μM myosinII and 50 μM blebbistatin[16,33], monomeric G-actin at a constant concentration of 10 μM was polymerized by adding one tenth of the sample volume of 250 mM imidazol, 40 mM MgCl, 10 mM EGTA, 250 mM KCl and 2 mM ATP. Samples were prepared at 4 °C to prevent polymerization before their location in a flow chamber.

To reduce surface interaction flow chambers were pretreated with $5 \, mg \, ml^{-1}$ casein. Samples were scanned with a 0.5 mW confocal blue (488 nm) argon laser spot of a commercial Leica TCS SP5 microscope at $31 \, cm \, s^{-1}$ for the number of n stimulation cycles (640 ms for one cycle). Once stimulated the network was imaged and followed in time using a red (633 nm) HeNe laser. This wavelength is too long to deactivate blebbistatin any further and a continuous sustained contraction was observable. Both for stimulation and subsequent observation a $20 \times$ (NA 0.7) oil immersion objective was used. Confocal timelapse $xyz$-stacks were obtained every 90 s to observe the dynamics of globally contracting gels. The projection of each $z$-stack is evaluated by line profiles and subsequent edge detection. The edges are not sharp and span a length of up to 20 μm. Samples were enclosed in sealed chambers to avoid any drift in the network. Addition of an ATP regeneration system, consisting of 20 mM creatine phosphate and $0.1 \, mg \, ml^{-1}$ creatine phosphokinase (Sigma-Aldrich) has no influence on the dynamics.

**Particle image velocimetry.** To investigate the internal dynamics 1 μm 'Fluoresbrite' microspheres (Polysciences) were added as tracer particles, and excited by a green (543 nm) HeNe laser. The displacement field was subsequently calculated by a custom-written PIV program implemented as an 'ImageJ' (Tseng, 'ImageJ', http://rsb.info.nih.gov/ij) plugin by Tseng (Supplementary Note 4). The PIV was performed through an iterative scheme. In each iteration, the displacement was calculated by the normalized correlation coefficient algorithm, so that an individual interrogation window was compared with a larger searching window. The next iteration took into account the displacement field measured previously, so that a false correlation peak due to insufficient image features is avoided. The normalized cross-correlation also allowed us to define an arbitrary threshold to filter out low correlation values due to insufficient beads present in the window. The resulting final grid size for the displacement field was $10 \, \mu m \times 10 \, \mu m$, with 98 beads per interrogation window on average. The erroneous displacement vectors were filtered out by their low correlation value and replaced by the median value from the neighbouring vectors[34].

**Digital image analysis of orientations.** The orientation of actin structures in Fig. 4 is inferred over time. First the analysed images are cropped to be sure to test only for alignment of internal structures unbiased by the boundary itself. Then Fourier spectrum analysis, adapted from the 'ImageJ' routine written by Jean-Yves Tinevez (http://imagej.net/Directionality), is applied. Structures with a preferred orientation generate a periodic pattern at $+90°$ orientation in the Fourier transform of the image, compared with the direction of the objects in the input image. This plugin chops the image into square pieces, and computes their Fourier power spectra. The latter are analysed in polar coordinates, and the power is measured for each angle using the spatial filters proposed by Liu[35]. The plugin computes a distribution indicating the amount of structures aligned in a given direction. Images with completely isotropic content will give a flat histogram, whereas images in which there is a preferred orientation will give a histogram with a peak at that direction. The area under the histogram is normalized to binning size. For an isotropic system the distribution is constant at $\approx 0.044$, because it is normalized to the chosen binning size of 8°.

**Code availability.** The Supporting Software contains the code to calculate Fig. 1b. More efficient implementations for large systems are available from the author M.K. upon request. The displacement field was calculated from PIV data via 'ImageJ' as described in the Methods section.

**Data availability.** The authors declare that data supporting the findings of this study are available within the article and its Supplementary Information files. The data shown in Figs 1b,2,3 and 4 are available from the corresponding author A.R.B. upon request. The theory predictions shown in Fig. 1c can be reproduced with the Supplementary 'Mathematica' notebook and are also available from author M.K. upon request. The theory predictions shown in Figs 2 and 3 can be reproduced using the Supplementary Equations quoted in the figure captions.

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

## Acknowledgements

Research was supported by ERC-SelfOrg (M.S., F.C.K. and A.R.B.), partly by the SFB863 and the Nanosystems Initiative Munich. M.K. acknowledges support by the Swiss National Science Foundation through grant 200021_156106. A.R.B. acknowledges the hospitality of the Miller Institute for Basic Research in Science at the University of Berkeley.

## Author contributions

M.S., F.C.K. and A.R.B. designed research. M.S. performed experiments. M.K. developed the models and wrote the software. M.S. and M.K. analysed the data and created the figures. Authors jointly wrote the paper.

## Additional information

**Competing financial interests:** The authors declare no competing financial interests.

**How to cite this article**: Schuppler, M. *et al.* Boundaries steer the contraction of active gels. *Nat. Commun.* **7**, 13120 doi: 10.1038/ncomms13120 (2016).

