## [Peer Review File · Nature Communications]

Reviewer #1 (Remarks to the Author)

This manuscript demonstrates experimentally how contraction of active actin-myosin gels is regulated by geometry as a consequence of force balances at the boundaries. The experimental findings are reproduced in a minimal spring model. However, it is not clear what we learn from this combination of experiments and model given that the model is phenomenological and is not linked to any relevant properties on the molecular level. Furthermore, I have concerns about the interpretation of several of the findings (as detailed below), and overall it is difficult to judge the validity of the conclusions since there is a serious lack of detail about the methods, analysis and fitting parameters used.

Major comments

- The physical model appears to nicely describe the experimental data, but it is unclear what we learn. The fitting parameters are nowhere specified, it is unclear how many fitting parameters there even are, nor is it clear whether the fitting parameters are the same in all cases or how strong they vary. Moreover, the elements of model should be connected to the experimental system.
- The interpretation of the mechanism of blebbistatin inactivation is doubtful. The authors interpret the role of blebbistatin as follows: "Thereby the blebbistatin remains covalently attached at the myosin heads, and no further competitive inhibition of the once activated motor is possible" (first paragraph in results and discussion). This picture is in fact not strongly supported by reference 16, where it is shown that inactivated blebbistatin binds to proteins including a BSA coated surface, indicating rather unspecific binding. This does not imply that myosin cannot be inhibited any more. It rather raises questions whether blebbistatin could alter myosin or actin behavior or even crosslink actin.
- The authors should explain the details of the stimulation protocol: How long is one stimulation cycle? Which wavelength is used for activation? Is the so called final statement always taken after 45min? How do they define the boundary of the shapes? How do they control filament length? They should systematically give the size of the area, shape of the area and number of stimulation cycles.
- The authors conclude that contraction speed depends on geometry: "As a conclusion we find that v_{max} is governed by the number of contractile units per length and the velocity of contraction of each unit cell¹⁷." It is unclear what they mean by the phrases "contractile units" and "unit cell". Their formulation alludes to a molecular mechanism for which they do not have proof.
- For the data presented in Figure 2a-b, key information is lacking. First of all, PIV data should be shown. In Fig 2a: How large is the circle? It is not clear how long the stimulations are and how this compares to the contraction dynamics. It would be helpful to give the radius as a function of time. At which time point is the velocity typically maximal (in particular interesting regarding the long stimulation time)? Does the velocity increase during stimulation? Why is v_{max} chosen as readout? How does v_{max} compare to the average velocity? Over which time interval is v_{max} measured? Where is v_{max} measured in the square? Since the velocities vary, it is not fair to measure the final state after 45min (which appears to be the case), the final state might be the same but time to get there varies. How is the radius measured? Could different degrees of densification bias this measurement, eg. a bright/dense clusters might appear larger than a dimer cluster. How many experiments were performed per condition? (applies also to Fig. 2b)
- For the data presented in Figure 2c-d, again key information is lacking. What was the size of the stimulated region and number of stimulations? They use a square shape: where in the square do they measure the velocity? Fig 2c: Where is the middle of the square, when the velocity becomes 0? Does negative velocity mean that this is the velocity on the other side of the square? Why is the reduction of the square size not visible (the length over which the velocity is measured presumably decrease)? And in panel Fig 2c: how are the filament lengths reduced (capping protein conc), and how was the stimulation done?
- There are alternative interpretations possible of the data on dynamics of contraction. The authors claim: "Thus, the information of unbalanced forces, starting from the boundary, penetrates the network during the onset of contraction. The maximal velocity of the boundaries is reached when

all contractile units between them sense their presence". They mention tensile stresses but do not consider the role of crosslinking or densification of actin (and presumably myosin). An alternative interpretation can be that the velocity at the boundary is initially larger because actin is partially crosslinked by inactive myosin. This material gets then dragged in the bulk, which increases the velocity there. This would mean the velocity is mainly dependent on the actin density.

- The model predicts identical aspect ratios before and after contraction. From movie 2, it appears that the experimentally observed contraction is not finished at the end of the movie and that the aspect ratio is changing during the contraction (first faster contraction along the short axis, than contraction along the long axis.) Again, it would be important to know when they evaluate the shape to be "final" and how they define the border of the shape. They introduce damped springs to account for their observations around the aspect ratio, mainly the fact that the contraction speed in the one direction is influenced by the side length of the other side in the other direction. This seems arbitrary. An alternative interpretation could again be based on gel densification: the gel cannot condense infinitely, however the density depends on the constriction on both sides, which leads to this coupling.

- The observation of filament alignment with spatially anisotropic attachment is in principle interesting. However, the authors should show the network before activation. Due to the anisotropic geometry of the chamber, the network may well be pre-aligned. Furthermore, the authors should check in their other experiments for alignment. They suggest that alignment in the anisotropic situation is due to the free border. However, it is alternatively possible that alignment could be related to the size of the contracted area in comparison to the average filament length, i.e. more alignment when the size of the constricting area is large compared to the average filament length.

-

Minor comments

- In Fig 2 the units for velocity are confusing: a,b: $\mu\text{m/s}$, s: $\mu\text{m/min}$, d 1/s (rate).

- Positions and lengths: it would be more clear if they could give the length systematically as distance from the boundary.

- Why are v^x_{max} and v^y_{max} equal at around $800\mu\text{m}$, I though the width of x is $400\mu\text{m}$. Is this a typo? (Fig. 3d)

Reviewer #2 (Remarks to the Author)

The article by Schuppler et al reports on experiments and modeling of light-activated contraction of reconstituted acto-myosin gels. The authors have developed a protocol that allows them to stimulate myosin activity and associated gel contraction locally via UV light. This is achieved by first assembling the acto-myosin network in the presence of blebbistatin that inhibits myosin activity, and then inactivating blebbistatin (and therefore activating myosin) in regions of controlled geometry by shining UV light. This procedure provides a powerful tool for controlled in-vitro studies of the mechanics of actomyosin networks. While it has been known for some time that blebbistatin can be photoinactivated by UV light (Refs. 15 and 16), but to my knowledge this is the first time that this tool has been used systematically to study the mechanics of actomyosin networks in vitro. Using this technique, the authors demonstrate a remarkably simple relation between the shape of the activated region and the boundary conditions imposed on the network and the shape and structure of the activated region after relaxation. They also provide a simple physical model that reproduces many of the experimental observations.

This work is significant because it suggests that cells may regulate local shape changes by controlling the spatial distribution of activated myosins. Especially interesting is the connection demonstrated in Fig. 4 between the activity-induced alignment of the actin network and the symmetry breaking arising from boundary conditions, which may indeed suggest a mechanism for the formation of stress fibers. Additionally, the simple spring model presented by the authors provides a good fit to the experimentally measured relation between the geometries of the initial and the final structures. It also shows that the contraction dynamics is controlled by a single time scale. I think the experimental results are sufficiently novel and significant to warrant publication

in Nature Communication. I was somewhat less enthusiastic about the minimal model that seems constructed to fit the experiments, but does not provide a lot of insight on the mechanisms that may organize the network in a critically damped state. I am overall inclined towards recommending publication after the authors have satisfactorily addressed the questions below.

1. A good portion of the paper is devoted to the discussion of the model, but little is said about the relation of the model's parameter to properties of the actomyosin network, such as stiffness or relaxation times, many of which are known from independent work, including some by the authors. For instance, which properties of the actomyosin network control what the authors call the radius of the interaction region and the rate of propagation of force imbalance within the network? Can model parameters such as the ratio K of the spring coefficients and τ_L (or τ_R) be estimated in terms of physical properties of acto-myosin networks? How do such estimates compare to experimental observations? Can one vary network properties (connectivity, myosin concentration) and use such data to determine the dependence of the model parameters on the properties of the network?

2. Fig 2(a) shows that the maximum contraction speed measured at the boundary of the activated region grows with the number of stimulation cycles to a plateau value that presumably corresponds to the situation where all available motors are activated. Similarly the ratio of the final to the initial size of the activated region rapidly saturates as a function of the number of stimulation cycles. It seems, however, that the time scales for these two saturations are quite different. Can the author comment more explicitly on this difference and on which properties of the network and/or of the myosin motors may control this relaxation? The discussion on page 7 makes qualitative sense, but can it be made more precise? From Fig 2b that shows that the maximum speed grows linearly with the initial linear size of the activated region it is argued that such maximum speed depends linearly on the number of contractile units per unit length in the network. Could this be tested independently by probing networks with different myosin concentrations?

3. The authors describe the actomyosin network as a set of coupled critically damped harmonic oscillators. This is an interesting idea, but it seems dynamical mechanisms would be needed to drive the system towards such a critical state (see in fact ref. 18), while their model is entirely static. Ref 18 suggests that in kinetic networks connectivity may be a key parameter in driving the system to a critically damped state. While the authors show in Fig 4c that they can vary the connectivity of their network and that a decrease in connectivity qualitatively changes the response, it seems such variations could be studied more systematically. Specifically, does an increase in connectivity also result in a qualitative change in the network response? Finally, as the authors know, ref. 14 proposed that network connectivity as controlled by the addition of a controlled amount of crosslinking proteins may drive the network to a marginal critical state where motors can contract the network into disjoint clusters with power law distributions. Is fig. 4c intended to suggest that a marginal network of the type described in Ref 14 (although the authors do not report the properties of the fractured network) can also be achieved by tuning the properties of the connectivity of the actomyosin network in the absence of additional crosslinks? If so, do the finite size of the active network and the boundary condition play a role?

4. Fig. 1 What does the color in the experimental images represent? It seems that the authors use fluorescent tracer particles imaged by PIV to track network displacements (see Methods section - perhaps a sentence on this could be included in the main text), but what precisely is imaged in red and yellow in Fig. 1b? This does not seem to be stated in the caption nor in the text. Same question about Figs 4a and 4c. Fig. 4c suggests that yellow is activated myosin since this is known from previous work on myosin clustering, but it should be said in the manuscript.

5. Can the authors comment on whether their technique can be used in vivo?

6. A small point: in Fig 2(b) do the black circles correspond to circle diameter and the red squares

to square median? Please indicate in the caption.

Reviewer #3 (Remarks to the Author)

The authors describe the use of light inactivation of blebbistatin to study the behavior of reconstituted networks of actin, crosslinkers and myosin motors. By shining light in defined regions, the authors are able to activate myosin in a spatially patterned way to induce contraction. The authors vary the size and shape of the activated region and the extent of activation (through the number of scan repetitions) and follow the contraction over time. They show that the dynamics of contraction and the shape evolution of the activated region are nicely captured by a simple model of initially relaxed overdamped springs whose spring constant depends on the local level of myosin activation. This model illustrates how tension propagates in the system, and how the evolution of the local contraction rate depends on the distance from the boundary. This also implies that the contraction rate along the different axes in an asymmetric shape will vary, and induces coupling between the principal axes of the system. Interestingly, this lead implies local alignment of the network springs in case of an initially asymmetric shape. This is demonstrated experimentally by following the contraction of a region with high aspect ratio.

Overall this is a nice paper which is presented in a clear manner. My only comment is that the implication of this work to cells and tissues is somewhat overstated in the manuscript. In cells both the boundary conditions and the network are expected to be more dynamic due to the turnover of the network and its components. Thus, the contractile behavior of cells is not expected to be as predetermined by the boundary conditions and observed the in vitro experiments presented in this manuscript. The authors should mention this in the discussion and tone down the claims.

Reviewer #1

This manuscript demonstrates experimentally how contraction of active actin-myosin gels is regulated by geometry as a consequence of force balances at the boundaries. The experimental findings are reproduced in a minimal spring model. However, it is not clear what we learn from this combination of experiments and model given that the model is phenomenological and is not linked to any relevant properties on the molecular level. Furthermore, I have concerns about the interpretation of several of the findings (as detailed below), and overall it is difficult to judge the validity of the conclusions since there is a serious lack of detail about the methods, analysis and fitting parameters used.

We thank the referee for his/her constructive criticism and are happy to address all raised points in detail below. We now much better emphasize the link between measurable quantities and model and added details about the methods, analysis and fitting parameters in an extended Supplementary Information (sections SI-5 and SI-3). All experimental details and fitting parameters are now also be found in the figure captions. As a matter of fact, we fully agree with the referee that in the original manuscript we unfortunately left out quite a number of details of the experiments and model due to aim of conciseness and we are happy to fill them in in the new version.

Major comments

- The physical model appears to nicely describe the experimental data, but it is unclear what we learn. The fitting parameters are nowhere specified, it is unclear how many fitting parameters there even are, nor is it clear whether the fitting parameters are the same in all cases or how strong they vary. Moreover, the elements of model should be connected to the experimental system.

All fitting parameters are now specified in the new Supplementary Information SI-5 and new Fig. S3 that shows values for model parameters K (ratio of spring coefficients) and τ (characteristic time of contraction) of rectangles with varying axis ratio. For the various geometries shown in Fig. 1 the model parameter τ does not enter, because we compare stationary results, and the only model parameter K is kept fixed. Within very limited bounds the model parameter K is a constant for all sets of experiments with filled circles and rectangles, as expected, because in the experiments we kept the stimulation density approximately constant while only varying the aspect ratio and area. Details about the experimental conditions are now presented in the figure captions and methods section.

Concerning the model parameter τ we now present this fitting parameter together with its expected behavior as function of axis ratio (Fig. S3a). This leaves us with just two fitting parameters, K and τ_{square} (for a square) that are used to describe the full dynamical behavior of various geometries at the chosen stimulation density. Moreover we have established a dependency of K on the number stimulation cycles (equation S1), and a relationship between τ_{square} , K , and the maximum velocity divided by the distance

between boundaries (equation S10, Figure 2b) enabling us to predict the full dynamical behavior of arbitrarily shaped geometries at arbitrary stimulation densities, provided we are still operating with the same 'material'. The effect of concentration, temperature etc. on the two model parameters we did not investigate but there is no reason to believe that more than two model parameters will be needed under different circumstances.

There is no microscopic theory for the two model parameters available, but they can be obtained from a single experiment for each new 'material' in the regime of fully saturated stimulation. See also answer to point 1 of referee #2 where we relate K to the elasticity ratio between active and inactive gels.

- The interpretation of the mechanism of blebbistatin inactivation is doubtful. The authors interpret the role of blebbistatin as follows: "Thereby the blebbistatin remains covalently attached at the myosin heads, and no further competitive inhibition of the once activated motor is possible" (first paragraph in results and discussion). This picture is in fact not strongly supported by reference 16, where it is shown that inactivated blebbistatin binds to proteins including a BSA coated surface, indicating rather unspecific binding. This does not imply that myosin cannot be inhibited any more. It rather raises questions whether blebbistatin could alter myosin or actin behavior or even crosslink actin.

The experimental results show unambiguously that the blebbistatin inactivation is indeed stable and a reliable method of activating the myosin activity. Thereby the exact microscopic mechanism is beyond the scope of the manuscript.

First of all, if the contraction would be stopped by the influx of unflashed blebbistatin, this should take place right after switching off the stimulation light, or within the diffusion time, which would be of 1 minute timescale. This is not the case, on the contrary, in many cases like asymmetric patterns or low motor density the velocity of the boundaries even get increased subsequent to terminated activation. This would not be the case, if inactivated blebbistatin was replaced by the remaining free and active blebbistatin of the solution. The time it takes for a molecule with the size of blebbistatin to get to the center of the activated pattern, according to basic diffusion equations, should be on the order of 1 minute. Contraction, however, continues for a period on the order of an hour. Importantly, we measure a dose dependence of motor activation by light for contractile activity, see Fig. 2a, and the model fits without any further assumption. This corresponds to an increased inactivation and stable inactivation of blebbistatin. This is why we can interpret the light reaction as covalent modification of the present proteins. We cannot exclude, that blebbistatin gets covalently attached to actin or even acts as crosslink. For the findings of this manuscript these possibilities would not change the further interpretation. A more microscopic picture can be obtained by the accompanying submitted paper by Murrell et al. It is reassuring to see that parts of the results we obtain in our experiments are readily reproduced in theirs, too – although on a slightly different geometry and length scale.

We now emphasise better on the experimental protocol of the photo-activation of the myosin in the beginning of the result section, the figure caption and also in the methods part.

- *The authors should explain the details of the stimulation protocol: How long is one stimulation cycle? Which wavelength is used for activation? Is the so called final statement always taken after 45min? How do they define the boundary of the shapes? How do they control filament length? They should systematically give the size of the area, shape of the area and number of stimulation cycles.*

We appreciate the question about additional details. We added them to the revised Methods section or the Supplementary Information SI-5, when appropriate.

The measurement time depends on motor density, 45 min is a typical time. The 'true' relaxed extension of the active gel is obtained by fitting an exponential function to the experimental results (equation S2). The boundary is measured putting a line profile and a subsequent edge detection. The filament length is not controlled and will be exponentially distributed. We wait until actin is polymerized sufficiently (approximately 10 min. after initiation of polymerization). The conditions for all essays except Fig. 2c,d and 5 are identical.

In Fig. 5 we demonstrate the influence of the presence of a capping protein.

The sizes of the areas and number of stimulation cycles are now presented in the figure captions.

- *The authors conclude that contraction speed depends on geometry: "As a conclusion we find that v_{max} is governed by the number of contractile units per length and the velocity of contraction of each unit cell17." It is unclear what they mean by the phrases "contractile units" and "unit cell". Their formulation alludes to a molecular mechanism for which they do not have proof.*

We do show that v_{max} depends on the length of the structures, which is then modeled by a number of springs, which are the contractile units. We do not want to allure to a molecular mechanism, but rather to a rather abstract modelling of the molecular details. This wording was actually introduced by Lenz *et al.* in Reference 17 and with the linear dependence of v_{max} on the initial diameter we refer to this concept and strengthen it. Importantly Murrell *et al* in the accompanying manuscript concentrate more on this aspect of contractile units.

- *For the data presented in Figure 2a-b, key information is lacking. First of all, PIV data should be shown.*

We now show PIV data corresponding to Fig. 2c in the Supplementary Fig. S4.

- *In Fig 2a: How large is the circle?*

The diameter of the circle (775 μm) is now mentioned explicitly within the revised manuscript (page 6).

- *It is not clear how long the stimulations are and how this compares to the contraction dynamics.*

Stimulations are always less than 1.5% of the observation time. Now mentioned in the Methods section.

- It would be helpful to give the radius as a function of time.

The radius of the stimulated circle as function of time is now given in the inset of 2b.

- At which time point is the velocity typically maximal (in particular interesting regarding the long stimulation time)?

Importantly, the stimulation was only done shortly at the beginning and the contraction happens without further activation. The characteristic point in time where the velocity is maximal is obtained by a fit and is identical with the model parameter, τ , as defined in equation S8. It thus depends on the axis ratio, in particular, and the motor density.

We improved the corresponding discussion in Supplementary Section SI-3. We now indicate the time point in the new inset of Fig. 2b.

- Does the velocity increase during stimulation?

Indeed, the velocity increases during stimulation, but since the stimulation time (640 ms) makes only a small fraction of the characteristic time (10-15 minutes) it can be ignored in the analysis. Added to the Methods Section.

- Why is v_{\max} chosen as readout? How does v_{\max} compare to the average velocity?

v_{\max} is chosen as readout because it is one of the clear signatures of the velocity $v(t)$, especially for the longer axis (x) of a rectangle, and because the time at which the maximum velocity is reached corresponds to one of the two model parameters, τ , that enters the long time behavior as well (equation S2). To test this equation, we use equation S6 to fit the experimental $v(t)$ curves. In Fig. 3a we indicate the position – it is clearly visible that the velocity is changing in time and first increases and then decreases again.

The ratio between average velocity and maximum velocity is readily calculated from the temporal evolution $v(t)$ given by equation S6, and v_{\max} given by equation S8. Combining the two equations gives the average velocity in terms of observation time T ,

$$\frac{\langle v_L \rangle(T)}{v_{\max}^L} = \frac{1}{v_{\max}^L T} \int_0^T v_L(t) dt = \left[\frac{\tau_L}{T} - \left(1 + \frac{\tau_L}{T} \right) e^{-T/\tau_L} \right] e$$

where $e \approx 2.718$. This ratio goes through a maximum at about $T \approx 5\tau_L / 9$, and ultimately approaches zero at $T \gg \tau_L$. The maximum value of the ratio is ≈ 0.81 . Using a typical $\tau_L = 10$ min., the ratio is close to 0.5 at $T = 45$ min. We have not added the mean velocity as function of observation time in the Supplementary Information because we are unsure about its relevance and because it can be readily calculated from equations S6 and S8.

- Over which time interval is v_{\max} measured? Where is v_{\max} measured in the square?

The maximum velocity v_{\max} is measured from the recorded time-dependent position of the boundary, $L(t)$, both for circles and squares. While $L(t)$ stands for a radius for the case of a circle, it stands for the median for the case of the square. While the direct evaluation of local slopes of $L(t)$ is prone to large error bars, we first fitted the measured $L(t)$ by equation S2 without assuming that there is an arbitrary coefficient in front of the t/τ_L term in the second bracket. This corresponds to the more general case of non-critical damping, and excellently captures our data with a coefficient close to unity. The analytical expression exhibits a maximum that is analytically expressed in terms of the fitting parameters. Insofar do we calculate v_{\max} with a small error from the full $L(t)$ curve. This is now more clearly mentioned in the Supplementary Information section SI-5.

- Since the velocities vary, it is not fair to measure the final state after 45min (which appears to be the case), the final state might be the same but time to get there varies.

We do not measure the final state after a fixed amount of 45 min. The remaining relaxation process is captured by assuming that equation S2 continues to hold after the measurement time has exceeded. This way the estimated final extension does not depend on the measurement time (added to Supplementary Section SI-5). Concerning Fig. 1 the shown states are close to equilibrium after 45 min., but not 100% equilibrated. A not fully equilibrated configuration would correspond in the model to a spring constant K that is not identical with the effective K .

- How is the radius measured?

The radius is measured using a line profile and subsequent edge detection, and the boundary line is located where the intensity is intermediate between active and passive regions.

- Could different degrees of densification bias this measurement, eg. a bright/dense clusters might appear larger than a dimer cluster.

We determine the sizes of the patterns with edge detection, and identify the mid point of the contrast change. The intensity decay is smooth and extends up to 10 microns. The error of the detection of the mid-point is therefore consistent throughout the experimental analysis. Only in cases where an oversaturation of the pixels would occur, a small change in the edge detection was observable. In the manuscript we have only one data point where a slightly bigger error occurred (+- 10 microns) which also results in a higher error in the $R_{\text{fin}}/R_{\text{init}}$ (last red data point in the Fig 2a). Therefore the error bars reflect the uncertainty in our analysis correctly. Details have been added to the Methods section.

- How many experiments were performed per condition? (applies also to Fig. 2b)

We performed a series of 2-5 experiments for each geometry. Each data point represents an individual experiment. Importantly, the whole series of sizes is performed with one batch of actin and myosin to exclude any batch-to batch variation and to capture the observed trends and dependencies. Repetition with a different batch results

in almost same quantitative data, but can be offset a bit, which makes an averaging not useful to extract the underlying physics.

- For the data presented in Figure 2c-d, again key information is lacking. What was the size of the stimulated region and number of stimulations?

The sizes of stimulated regions and number of stimulations are now collected for all reported data in Tab. S1 of the extended Supplementary Information. The initial median for the data shown in Figs. 2c-d was 581 μm .

- They use a square shape: where in the square do they measure the velocity?

The position-dependent velocity and contraction rate is measured at the median of the square. We now added a corresponding comment, and PIV data, shown by Figure 4.

- Fig 2c: Where is the middle of the square, when the velocity becomes 0?

The middle of the patterns does not remain exactly in the center of the recorded images, while time proceeds. During the measurement time of 90 min. the center-of-mass position typically varies not more than by 40 μm . This overall “small” center-of-mass motion does not affect any of the measured quantities as we determine the location of the boundary line. In case of the square evaluated in Fig. 2c the movement of the center is 22 μm . Quantities shown in Fig. 2d at fixed distance to the boundary are then calculated within the coordinate frame given by the boundary line. This is now mentioned in the Supplementary Information. The velocity becomes zero in the middle of the square. For the data shown in Fig. 2c the middle of the square is at ‘position’ $(581/2) \mu\text{m} = 290 \mu\text{m}$.

- Does negative velocity mean that this is the velocity on the other side of the square?

Yes. To be able to show velocities for the whole stimulated region velocities are shown using a fixed coordinate system, while velocities are considered positive (inward radial velocity for a circle) to simplify the following discussion.

- Why is the reduction of the square size not visible (the length over which the velocity is measured presumably decrease)?

Indeed Figure 2 was misleading, as it is a perspective plot of the velocity and space – this leads to a distortion and gives a wrong impression how the length evolves. We now improved the figure 2c and 2d and figure caption and hope that it now demonstrates better the point we make.

- And in panel Fig 2c: how are the filament lengths reduced (capping protein conc), and how was the stimulation done?

The filament lengths were not reduced in Fig. 2c. The only experimental data shown with reduced average initial lengths is Fig. 4c. We reduced the filament length by the addition of 2 nM capping protein. The stimulation was in this case $n = 20$ cycles. Details about the filament length adjustment, stimulation procedure and stimulation times are now provided in the figure captions.

- There are alternative interpretations possible of the data on dynamics of contraction. The authors claim: "Thus, the information of unbalanced forces, starting from the boundary, penetrates the network during the onset of contraction. The maximal velocity of the boundaries is reached when all contractile units between them sense their presence". They mention tensile stresses but do not consider the role of crosslinking or densification of actin (and presumably myosin). An alternative interpretation can be that the velocity at the boundary is initially larger because actin is partially crosslinked by inactive myosin. This material gets then dragged in the bulk, which increases the velocity there. This would mean the velocity is mainly dependent on the actin density.

Fig. R1 (not part of the revised manuscript): Actin is not redistributed in the contracting network.

Only the relative distances between inhomogeneities in the contractile network are changed during contraction. We exemplify this by plotting above (Fig. R1) the intensity profiles through a contracting network and indicate the position of prominent structure. Clearly the structure do not change their positions, the intensities remain comparable.

We do not fully understand the suggested "alternative interpretation". We do not observe a heterogeneous distribution of material through the activated area at the beginning directly after the activation. Maybe it is important to stress out, that the activated area is embedded in a comparable huge environment, which is the reason why we see completely homogenous and isotropic distribution of the constituents, right before and right after stimulation.

The movies and the displacement fields show unambiguously, that there is no positional reorganization happening during the contraction. The network is contracting like fish net

– only the relative distance between points change due to the contraction activity. This is the affine deformation we are discussing in the manuscript. This gives rise to the interpretation we are presenting here. This deformation is contrary to a steady state where continuous rupture and net transport happens.

- The model predicts identical aspect ratios before and after contraction. From movie 2, it appears that the experimentally observed contraction is not finished at the end of the movie and that the aspect ratio is changing during the contraction (first faster contraction along the short axis, then contraction along the long axis.) Again, it would be important to know when they evaluate the shape to be "final" and how they define the border of the shape.

We fully agree with the statement of the referee. In general, the final values for shape and axis ratio are obtained from the fitted $L(t)$ curves, and not from the transient values at the end of the observation period. While the aspect ratio is going through a maximum in the course of time, the final axis ratio $X_{\text{fin}}/Y_{\text{fin}}$ becomes identical with the ratio of initial axis ratios $X_{\text{init}}/Y_{\text{init}}$. This is actually one of our major findings (equation S3): $L_{\text{fin}}/L_{\text{init}} = K$ for $L=X$ and $L=Y$, and it implies $X_{\text{fin}}/Y_{\text{fin}} = X_{\text{init}}/Y_{\text{init}}$. This is highlighted in the manuscript and Supplementary Section SI-3. The procedures to evaluate the interfacial border as well as L_{init} and L_{fin} had been mentioned above, and they entered Supplementary Section SI-5.

- They introduce damped springs to account for their observations around the aspect ratio, mainly the fact that the contraction speed in the one direction is influenced by the side length of the other side in the other direction. This seems arbitrary. An alternative interpretation could again be based on gel densification: the gel cannot condense infinitely, however the density depends on the constriction on both sides, which leads to this coupling.

The referee is right that alternative modeling approaches are possible, yet we decided to stick to the simplest assumptions. The springs must not be linear, and we could introduce excluded volume interactions to make the model more complicated and nonlinear. The mismatch of spring coefficients leads to gel densification in our model. Under all experimental conditions we find that $X_{\text{fin}}/Y_{\text{fin}} = X_{\text{init}}/Y_{\text{init}}$ holds. This actually proves in our eyes that excluded volume interactions are not relevant. The reason for presenting the model with linear springs is that linear springs are already sufficient to describe both the static and dynamical behavior of these stimulated gels. The observed critically damped oscillator behavior is incompatible with significantly nonlinear springs and the identical axis ratios are incompatible with excluded volume interactions. For the case of linear springs the final configuration can be calculated trivially from the initial one (Supplementary Code), while nonlinear springs and other gel densification mechanism would not only require a numerical implementation, but also introduce complexity and additional model parameters without any gain in our understanding.

- The observation of filament alignment with spatially anisotropic attachment is in principle interesting. However, the authors should show the network before activation. Due to the anisotropic geometry of the chamber, the network may well be pre-aligned. Furthermore, the authors should check in their other experiments for alignment. They suggest that alignment in the anisotropic situation is due to the free border. However, it

is alternatively possible that alignment could be related to the size of the contracted area in comparison to the average filament length, i.e. more alignment when the size of the constricting area is large compared to the average filament length.

The referee is absolutely right, that it is important to present a proof of the isotropic starting condition. We now infer the orientation of actin structures in the channel over time not only visually, but quantitatively by means of an FFT image analysis, see revised Fig. 4b and Methods section.

For the other experiments, it is important to realize, that the stimulated areas are significantly bigger than the filament length or mesh size of the network. At the same time the activated area is almost an order of magnitude smaller than the total sample size. Filling of the chambers was done before polymerization and with extreme care to prevent prealignment. Any prealignment would result in non-isotropic and especially non-consistent contractions.

Minor comments

- In Fig 2 the units for velocity are confusing: a,b: $\mu\text{m/s}$, s: $\mu\text{m}/\text{min}$, d 1/s (rate).

We now use μm and min for all plots.

- Positions and lengths: it would be more clear if they could give the length systematically as distance from the boundary.

Throughout the manuscript we extensively evaluate velocities of the boundaries as the significant quantity revealing the discussed mechanisms. The only plot, where the distance to boundary over time is important is Fig. 2d, where the distance to boundary is highlighted with a color coded frame.

- Why are v^x_{max} and v^y_{max} equal at around $800\mu\text{m}$, I thought the width of x is $400\mu\text{m}$. Is this a typo? (Fig. 3d)

This is not a typo, it is correct. We kept on using Xinit as in the other subfigures for consistency, and stated that Xinit= $400\mu\text{m}$ is the half length.

Reviewer #2

The article by Schuppler et al reports on experiments and modeling of light-activated contraction of reconstituted acto-myosin gels. The authors have developed a protocol that allows them to stimulate myosin activity and associated gel contraction locally via UV light. This is achieved by first assembling the acto-myosin network in the presence of blebbistatin that inhibits myosin activity, and then inactivating blebbistatin (and therefore activating myosin) in regions of controlled geometry by shining UV light. This procedure provides a powerful tool for controlled in-vitro studies of the mechanics of actomyosin networks. While it has been known for some time that blebbistatin can be photoinactivated by UV light (Refs. 15 and 16), but to my knowledge this is the first time that this tool has been used systematically to study the mechanics of actomyosin

networks in vitro. Using this technique, the authors demonstrate a remarkably simple relation between the shape of the activated region and the boundary conditions imposed on the network and the shape and structure of the activated region after relaxation. They also provide a simple physical model that reproduces many of the experimental observations.

We thank this referee for the encouraging assessment of our efforts.

This work is significant because it suggests that cells may regulate local shape changes by controlling the spatial distribution of activated myosins. Especially interesting is the connection demonstrated in Fig. 4 between the activity-induced alignment of the actin network and the symmetry breaking arising from boundary conditions, which may indeed suggest a mechanism for the formation of stress fibers. Additionally, the simple spring model presented by the authors provides a good fit to the experimentally measured relation between the geometries of the initial and the final structures. It also shows that the contraction dynamics is controlled by a single time scale. I think the experimental results are sufficiently novel and significant to warrant publication in Nature Communication. I was somewhat less enthusiastic about the minimal model that seems constructed to fit the experiments, but does not provide a lot of insight on the mechanisms that may organize the network in a critically damped state. I am overall inclined towards recommending publication after the authors have satisfactorily addressed the questions below.

We thank him/her also for stressing the significance, and are going to address all six questions below.

1. A good portion of the paper is devoted to the discussion of the model, but little is said about the relation of the model's parameter to properties of the actomyosin network, such as stiffness or relaxation times, many of which are known from independent work, including some by the authors. For instance, which properties of the actomyosin network control what the authors call the radius of the interaction region and the rate of propagation of force imbalance within the network? Can model parameters such as the ratio K of the spring coefficients and τ_L (or τ_R) be estimated in terms of physical properties of acto-myosin networks? How do such estimates compare to experimental observations? Can one vary network properties (connectivity, myosin concentration) and use such data to determine the dependence of the model parameters on the properties of the network?

We fully agree with this referee that the original manuscript did not provide enough relations between the parameters and the known parameters of the system. We improved this now by incorporating a more quantitative discussion of the dependencies in Supplementary Section SI-3 and added Figure S3. Yet, it is important to stress out that the aim is not to present a microscopic interpretation, but a coarse grained simple model capturing the observations and revealing the underlying governing physical principles.

The two model parameters are experimentally related to the physical system by the fit of the effect of number of stimulations on the ratio of spring coefficients, K , and by

providing a dependency of the characteristic contraction time on the axis ratio. We have replaced ‘relaxation time’ by ‘contraction time’ throughout the manuscript, as this characteristic time is not directly related to the dynamical properties of the inactive gel, but rather a measure for the contractile activity.

The ratio K is a ratio of spring constants of the model, and it turns out that the found values would imply a 4 to 6 times higher modulus of the active gel compared to the passive one. This is not an unreasonable value. Yet, such a direct comparison of the ratio K with the ratio of modules is dangerous, and even misleading. In the experimental system the active region has a modul, and an embedded active force.

The embedded force can lead to an increase of the modul, but just an increase of the module alone would not lead to a contraction. It is the pre-stressed form of the gel which come from the stimulation and is given by the boundary condition, which subsequently induces the contraction. At the end it is the active force of the motors acting on the module (network), which leads to the contraction.

In the simulation the contraction can be included by the either a different spring constant or a different equilibrium length of the springs in the active area. For the simulation the former was used. In a real system an intricated interplay between contraction and increase of modul happens. The more it is suprising, that such a simple model fully captures the static and dynamic properties of the system.

2a. Fig 2(a) shows that the maximum contraction speed measured at the boundary of the activated region grows with the number of stimulation cycles to a plateau value that presumably corresponds to the situation where all available motors are activated. Similarly the ratio of the final to the initial size of the activated region rapidly saturates as a function of the number of stimulation cycles. It seems, however, that the time scales for these two saturations are quite different. Can the author comment more explicitly on this difference and on which properties of the network and/or of the myosin motors may control this relaxation?

We understand the worries of the referee. By eye it looks as if the v_{\max} curve ‘relaxes’ faster in Fig. 2a. Because the time between subsequent stimulations (n) does not affect our results, there is no time scale in our plot “versus stimulation cycles”. Stimulation takes place within maximal 1.7 min for $n=160$ while contraction appears over 90 min. The number of stimulations is rather related, but not fully proportional, to the density of active motors. The theory expression used for v_{\max} in terms of n is equation S12, the one for K in terms of n is equation S1, as we mention now in the caption of Fig. 2a. Both v_{\max} and K thus depend on the actual value of n only, there is no delay or relaxation. The apparent delay is only a visual impression one encounters when comparing $K(n)$ with $[1/K(n)] - 1$. According to our equation S1 there are two quantities that determine $K(n)$. (i) The ratio of spring coefficients in the saturated case, K_{∞} , and (ii) the stimulation efficiency, P , that corresponds to the fraction of newly stimulated material during a single stimulation. Obviously, both quantities depend on the chosen experimental conditions (material, laser), but if the conditions are kept, the first two measurements are sufficient to estimate both K_{∞} and P for all subsequent measurements of various

geometries at arbitrary amounts of stimulation. To summarize, there is no difference in time scales, and for the effect of n on K and v_{\max} we provide analytic expressions.

2b. The discussion on page 7 makes qualitative sense, but can it be made more precise? From Fig 2b that shows that the maximum speed grows linearly with the initial linear size of the activated region it is argued that such maximum speed depends linearly on the number of contractile units per unit length in the network. Could this be tested independently by probing networks with different myosin concentrations?

We agree with the referee that we did not make enough clear how we test this qualitative argument. We have added experimental evidence for the statements made to this paragraph. Fig. 2b shows that v_{\max} grows linearly with the initial linear size while the density of contractile units remains unchanged (identical n). According to equation S8 the maximum velocity is not only proportional to the initial linear size, but also linear in K . Because K is basically the spring coefficient of the active gel, it is plausible to assume that it grows linearly with the density of contractile units. This could indeed be tested by probing networks with different myosin concentrations, after a fixed amount of stimulation cycles. Yet, we decided to vary the activity by the number of stimulation cycles, and prefer to vary the activity by these means as this ensures the most sensitive and best quantitative reproducibility of the experiments. The fact that the observed $K(n)$ matches the predicted $K(n)$ upon assuming that K is proportional to the motor density can be considered as a confirmation of our claim. See revised caption of Fig. 2.

3. The authors describe the actomyosin network as a set of coupled critically damped harmonic oscillators. This is an interesting idea, but it seems dynamical mechanisms would be needed to drive the system towards such a critical state (see in fact ref. 18), while their model is entirely static. Ref 18 suggests that in kinetic networks connectivity may be a key parameter in driving the system to a critically damped state. While the authors show in Fig 4c that they can vary the connectivity of their network and that a decrease in connectivity qualitatively changes the response, it seems such variations could be studied more systematically. Specifically, does an increase in connectivity also result in a qualitative change in the network response? Finally, as the authors know, ref. 14 proposed that network connectivity as controlled by the addition of a controlled amount of crosslinking proteins may drive the network to a marginal critical state where motors can contract the network into disjoint clusters with power law distributions. Is fig. 4c intended to suggest that a marginal network of the type described in Ref 14 (although the authors do not report the properties of the fractured network) can also be achieved by tuning the properties of the connectivity of the actomyosin network in the absence of additional crosslinks? If so, do the finite size of the active network and the boundary condition play a role?

We discuss the stationary solution of our dynamical equations in Fig. 1 to compare with experimental data, but our model is not a static one, it is a dynamical model, where forces arise due to springs and damping. The conditions under which a complex (multi-spring) system exhibits critically damped dynamics is a question that remains so far mostly unsolved, and is under study by several groups, mainly mathematicians. We are

not in the position to answer this question as a side-result of the present work. The referee is right that our Ref. 18 suggests that in kinetic network models connectivity may play a role. All presented experimental data with the exception of Fig. 4c (right) concerns systems that we believe are fully, and homogeneously connected. Fig. 4c (right) we have added to clarify, that network connectivity may play a role, but the present work did not investigate the effect of variation of the initial average F-actin length. We find that the system exhibits critically damped dynamics if the F-actin lengths are sufficiently large to ensure a homogeneously connected network. We do not know quantitatively below which F-actin length this property disappears, but all results presented and discussed in this manuscript (see Fig. 1 in particular) provide evidence that the connectivity must be preserved during stimulation. A stimulated gel that gets disconnected from the surrounding inactive gel would not evolve into a concave structure (cf Fig. 1b-2), but into a convex one. Network connectivity as controlled by the addition of a controlled amount of crosslinking proteins may drive the network to a marginal critical state where motors can contract the network into disjoint clusters.

This scenario is not observed in the experiments leading to Figs. 1-3, but the mechanism could be of relevance for interpreting Fig. 4c-right, and is therefore mentioned (Fig. 5). The effect of network connectivity of elastic Lennard-Jones networks that differ from our model by additional attractive short-ranged interactions had been elucidated in previous works, now mentioned in the caption of Fig. 5.

Indeed, in our opinion, our experimental system might be suited to study connectivity of a prepolymerized actin system further. We can tune connectivity by changing initial actin length distribution by capping protein. Ref. 14 needed crosslinks because motors started pulling on actin from the very beginning of polymerization. Here the blebbistatin bound myosin acts as a crosslinker and no active motors are present without activation.

We indeed think the finite size and the boundary condition do play a role, since in the disrupting case (e.g. Fig. 4c) a large share of the active material gets pulled towards the boundary. Then for evaluation, it is hard to pinpoint a final state, and consequently a final size of the clusters as they (slowly) drag in more and more material from the passive areas. A consequence for the setup is that you have to activate the whole gel and carefully passivate the boundaries (as done by Ref.14) It is not trivial to passivate the interaction of a gel of this initial density and simultaneously reduce the sample size to the dimensions activatable with our laser. In our opinion, this would be a new though very interesting story, which we may pursue in near future.

4. Fig. 1 What does the color in the experimental images represent? It seems that the authors use fluorescent tracer particles imaged by PIV to track network displacements (see Methods section - perhaps a sentence on this could be included in the main text), but what precisely is imaged in red and yellow in Fig. 1b? This does not seem to be stated in the caption nor in the text. Same question about Figs 4a and 4c. Fig. 4c suggests that yellow is activated myosin since this is known from previous work on myosin clustering, but it should be said in the manuscript.

The color is fluorescence from the labeled actin – the activated regions are brighter due to the accumulation of actin induced by the contraction right after the activation. We

now explain this much better than we did prefer in the manuscript and the figure caption.

5. Can the authors comment on whether their technique can be used in vivo?

According to reference 16, the technique cannot be used in vivo. Yet there are attempts in chemistry groups to synthesis other derivatives of blebbistatin on their way. These would render the photo-activation less toxic and would enable such experiments, which would be really very nice to do.

We decided not to include such speculations into the manuscript.

6. A small point: in Fig 2(b) do the black circles correspond to circle diameter and the red squares to square median? Please indicate in the caption.

Done.

Reviewer #3

The authors describe the use of light inactivation of blebbistatin to study the behavior of reconstituted networks of actin, crosslinkers and myosin motors. By shining light in defined regions, the authors are able to activate myosin in a spatially patterned way to induce contraction. The authors vary the size and shape of the activated region and the extent of activation (through the number of scan repetitions) and follow the contraction over time. They show that the dynamics of contraction and the shape evolution of the activated region are nicely captured by a simple model of initially relaxed overdamped springs whose spring constant depends on the local level of myosin activation. This model illustrates how tension propagates in the system, and how the evolution of the local contraction rate depends on the distance from the boundary. This also implies that the contraction rate along the different axes in an asymmetric shape will vary, and induces coupling between the principal axes of the system. Interestingly, this lead implies local alignment of the network springs in case of an initially asymmetric shape. This is demonstrated experimentally by following the contraction of a region with high aspect ratio.

Overall this is a nice paper which is presented in a clear manner. My only comment is that the implication of this work to cells and tissues is somewhat overstated in the manuscript. In cells both the boundary conditions and the network are expected to be more dynamic due to the turnover of the network and its components. Thus, the contractile behavior of cells is not expected to be as predetermined by the boundary conditions and observed the in vitro experiments presented in this manuscript. The authors should mention this in the discussion and tone down the claims.

We thank the referee for recommending the manuscript for publication. The referee suggest to tone down the statements for the implications of cytoskeletal structures in cells. We fully agree that turnover and dynamics of the cell will alter the picture – yet as long as a structure will be load bearing the boundary effects described here will play a role. To our – probably limited – understanding of the current understanding of stress fiber formation or muscle development it is a mandatory prerequisite to have tension in the structure that the self organization takes place, which would argue that the symmetry break at the boundary translates into the tissue.

To not over-complicate matters too much, now we point out the difference between the static structures we have here and the importance of dynamic turnover in cells in the conclusion.

Reviewer #1 (Remarks to the Author)

The manuscript is substantially improved and the authors have handled most of the reviewers' comments quite well. There are a few remaining issues:

- the authors still do not comment very explicitly in their manuscript on the physical interpretation of the two model parameters that characterize the active networks, K and τ . They should discuss more explicitly the physical interpretation and point out that it will be interesting in future to connect these parameters to microscopic theories, referencing examples of existing microscopic theories/particle-based simulations of contractile networks.
- the authors should make clear that their statement that blebbistatin covalently modifies myosin is not something that is well-established, but rather it is something they conclude from their own data [working details of their response into the paper]
- the authors should comment explicitly in the manuscript on alternative interpretations of their data, in particular gel densification, and explain why they favor the critically damped oscillator model [again working details of their response into the paper]

Reviewer #2 (Remarks to the Author)

I think the authors have answered satisfactorily most (but not all, see below) of my questions. Overall I think the authors present an interesting controlled approach for probing the geometry of actomyosin network contractility, accompanied by a model that, while straightforward, does provide a simple interpretation of the data. Before publication, however, they need to address the point below.

Both I and Referee I had questions about the parameters of the model. I do not find the answers satisfactory. The authors point to the new section SI-5 for a summary of the fitting parameters, but I do not find here a clear list nor description.

Two models are presented: the network spring model and the minimal model where one only looks at deformations of the boundaries.

The description of the network spring model is confusing. It seems the only parameter in this model is the ratio K of the spring constants and the model is described as a static one, where one creates a force imbalance by instantaneously increasing the spring constants in a chosen region of the network and then solving for the resulting displacements by inverting linear equations. But both Fig S1 and the authors' answer to my point 3 indicate that this is actually a dynamical model with relaxational dynamics. If so, how is the dynamics implemented? Is there a characteristic time scale in the model making the total number of parameters at least two? How is this time scale related to the time scale that controls the relaxation in the minimal model? The relationship between the spring network and minimal model is then discussed in SI-4, but the relationship among the time scales is not clear to me.

The minimal model, if I understand it correctly, contains essentially two parameters: a relaxation time and the parameter K defined as the ratio of the initial to the final linear size of the activated region. Since in the network spring model this ratio is found to be determined entirely by the ratio of the spring constants, the parameter K in the minimal model is identified with the parameter K in the spring network model. Hence the suggestion that physically this parameter represents the ratio of the active to passive gel. This sounds plausible, but I did not find it stated in the text. The time scale depends on the shape of the region and for anisotropic shapes can be different in different directions, hence the notation τ_L , τ_X , τ_Y , etc. This should be stated clearly as now the notation is confusing and suggests a proliferation of time scales, while it seems there is effectively only 1. Then in section SI-5 they argue that τ can be obtained from experimental measurements and the only fitting parameter is K , but the reader has to work very hard to collect

all this information from different places in the text and SI. A clearer presentation is needed.

A smaller point: there are no units on the vertical axis of Fig. S-3. Are these minutes? What physical properties of the network would determine a relaxation time of minutes?

Reviewer #1

The manuscript is substantially improved and the authors have handled most of the reviewers' comments quite well. There are a few remaining issues:

We thank the referee for his/her positive assessment and remaining constructive criticism and are happy to address the points in detail below, and within the revised manuscript.

- The authors still do not comment very explicitly in their manuscript on the physical interpretation of the two model parameters that characterize the active networks, K and τ . They should discuss more explicitly the physical interpretation and point out that it will be interesting in future to connect these parameters to microscopic theories, referencing examples of existing microscopic theories/particle-based simulations of contractile networks.

Indeed, a microscopic interpretation is beyond the scope of the present manuscript and the parameters can now only be discussed as rather phenomenological parameters. While K can be easily understood as the elastic ratio of active/passive gel, the time constant is more difficult to imagine. We provide a well motivated expression for τ as function of axis ratio, this dependency should stay valid within a more microscopic theory. We now point out that future work needs to connect these parameters to microscopic mechanisms.

- The authors should make clear that their statement that blebbistatin covalently modifies myosin is not something that is well-established, but rather it is something they conclude from their own data [working details of their response into the paper]

We changed the 3rd sentence of the Results and Discussion Section to: “Thereby, we find that no further competitive inhibition of the once activated motor is possible by the remaining freely diffusing active blebbistatin in the surrounding. Thus, the sustained activity of the illuminated area over a time scale of an hour indicates that the inactivated blebbistatin remains covalently attached at the myosin heads [Ref 16].”

- The authors should comment explicitly in the manuscript on alternative interpretations of their data, in particular gel densification, and explain why they favor the critically damped oscillator model [again working details of their response into the paper]

We added the following sentence (highlighted in blue) on page 8 of the revised manuscript: “The maximal velocity of the boundaries is reached when all contractile units between them sense their presence. From tracing network inhomogeneities or fiducial tracer particles, we conclude that the network contracts affinely by reducing the relative distances without positional redistributions of material inside the network. The increase of the speed gradient correlates thereby with the densification of the network.”

Reviewer #2

I think the authors have answered satisfactorily most (but not all, see below) of my questions. Overall I think the authors present an interesting controlled approach for probing the geometry of actomyosin network contractility, accompanied by a model that, while straightforward, does provide a simple interpretation of the data. Before publication, however, they need to address the point below. Both I and Referee I had questions about the parameters of the model. I do not find the answers satisfactory. The authors point to the new section SI-5 for a summary of the fitting parameters, but I do not find here a clear list nor description.

The revised manuscript contains all fitting parameters in the figure captions. We have erased earlier section SI-4 and revised SI-3. New Figure S3 contains the fitting parameters for all rectangles. Please see also our response to Reviewer #1.

Two models are presented: the network spring model and the minimal model where one only looks at deformations of the boundaries. The description of the network spring model is confusing. It seems the only parameter in this model is the ratio K of the spring constants and the model is described as a static one, where one creates a force imbalance by instantaneously increasing the spring constants in a chosen region of the network and then solving for the resulting displacements by inverting linear equations. But both Fig S1 and the authors' answer to my point 3 indicate that this is actually a dynamical model with relaxational dynamics. If so, how is the dynamics implemented? Is there a characteristic time scale in the model making the total number of parameters at least two? How is this time scale related to the time scale that controls the relaxation in the minimal model? The relationship between the spring network and minimal model is then discussed in SI-4, but the relationship among the time scales is not clear to me.

We apologize for confusion we might have caused by our previous response letter and the supplement. The referee is correct: we are presenting two models, the static network spring model for the prediction of shape changes for arbitrary stimulated shapes (including their interfaces), and the minimal dynamic model for the dynamic behavior of the characteristic positions of the simple rectangular and circular interfaces. We have revised Fig S1, the Supplementary Sections, and manuscript. Panel (a) in Fig. S1 now describes the static network model with two types of strong and weak springs, a parameter K and no time scale, and panel (b) the minimal dynamic model with effective springs and dashpots, giving rise to the compression time τ_c . Although the models are mostly independent, the parameter K obtained by the static spring model appears in the minimal dynamic minimal model, as it captures the stationary state of the minimal dynamic model (see revised first paragraph of the Supplemental Section SI-3).

We adopted and rewrote parts of the manuscript and the supplement to clarify this point better.

The minimal model, if I understand it correctly, contains essentially two parameters: a relaxation time and the parameter K defined as the ratio of the initial to the final linear size of the activated region. Since in the network spring model this ratio is found to be determined entirely by the ratio of the spring constants, the parameter K in the minimal

model is identified with the parameter K in the spring network model. Hence the suggestion that physically this parameter represents the ratio of the active to passive gel. This sounds plausible, but I did not find it stated in the text. The time scale depends on the shape of the region and for anisotropic shapes can be difference in different directions, hence the notation τ_L , τ_X , τ_Y , etc. This should be stated clearly as now the notation is confusing and suggests a proliferation of time scales, while it seems there is effectively only 1. Then in section SI-5 they argue that τ can be obtained from experimental measurements and the only fitting parameter is K , but the reader has to work very hard to collect all this information from different places in the text and SI. A clearer presentation is needed.

The referee is absolutely correct, and we improved the text that all this can now be found in the main text (page 9, in particular, and revised first paragraph of the Supplemental Section SI-3). We eliminated τ_R and the function $\tau(a)$ from the manuscript, and instead introduced τ_c , the only relevant compression time for circles and squares, while the compression times for the two axes of a rectangle are identical with τ_c , multiplied by a factor that depends on axis ratio only. We added a sentence clarifying that there is only one characteristic time, and that the others derive from it. The τ_c can indeed be obtained from experimental measurement, most conveniently by measuring the radius of a circular interface as function of time, for a circle of arbitrary initial radius, as τ_c is insensitive to this initial radius. τ_c then corresponds to the time where the (inward) velocity of the interface goes through a maximum. If one subtracts the final value L_{fin} from the experimental data $L(t)$, and divides this difference by $L_{fin}-L_{init}$, all experimental data falls onto a single curve if the time axis is t/τ . This is all mentioned in the revised manuscript.

A smaller point: there are no units on the vertical axis of Fig. S-3. Are these minutes? What physical properties of the network would determine a relaxation time of minutes?

We thank the reviewer for this observation and added 'min' to the caption of Fig. S3. The compression time τ_c is determined by the densification and concomitant increase of the modulus of the gel and the finite speed and stall force of myosin II. We do not have a formula that expresses τ_c in terms of microscopic quantities, but we studied the effect of number of stimulation cycles on τ_c .